# Effects of Laser Power and Hatch Orientation on Final Properties of PA12 Parts Produced by Selective Laser Sintering

**DOI:** 10.3390/polym14173674

**Published:** 2022-09-04

**Authors:** Anouar El Magri, Salah Eddine Bencaid, Hamid Reza Vanaei, Sébastien Vaudreuil

**Affiliations:** 1Euromed Research Center, Euromed Polytechnic School, Euromed University of Fes, Route de Meknès (Rond point Bensouda), Fes 30 000, Morocco; 2Léonard de Vinci Pôle Universitaire, Research Center, 92916 Paris La Défense, France; 3Arts et Métiers Institute of Technology, CNAM, LIFSE, HESAM University, 75013 Paris, France

**Keywords:** selective laser sintering, PA12, laser power, hatch orientation, annealing

## Abstract

Poly(dodecano-12-lactam) (commercially known as polyamide “PA12”) is one of the most resourceful materials used in the selective laser sintering (SLS) process due to its chemical and physical properties. The present work examined the influence of two SLS parameters, namely, laser power and hatch orientation, on the tensile, structural, thermal, and morphological properties of the fabricated PA12 parts. The main objective was to evaluate the suitable laser power and hatching orientation with respect to obtaining better final properties. PA12 powders and SLS-printed parts were assessed through their particle size distributions, X-ray diffraction (XRD), Fourier Transform Infrared spectroscopy (FTIR), differential scanning calorimetry (DSC), a scanning electron microscope (SEM), and their tensile properties. The results showed that the significant impact of the laser power while hatching is almost unnoticeable when using a high laser power. A more significant condition of the mechanical properties is the uniformity of the powder bed temperature. Optimum factor levels were achieved at 95% laser power and parallel/perpendicular hatching. Parts produced with the optimized SLS parameters were then subjected to an annealing treatment to induce a relaxation of the residual stress and to enhance the crystallinity. The results showed that annealing the SLS parts at 170 °C for 6 h significantly improved the thermal, structural, and tensile properties of 3D-printed PA12 parts.

## 1. Introduction

Additive manufacturing (AM), also known as 3D printing or direct digital manufacturing, has become an alternative technology that competes with more mature technologies such as casting and forging in different industrial fields including aerospace, automotive, and biomedical fields [1,2,3,4]. It is defined as the process of joining materials to create objects from 3D model data, usually layer upon layer, as opposed to subtractive manufacturing methodologies [5]. One of the major benefits of this promising technology is the freedom of its design and its facilitation of the printing of complex geometries. It gives engineers and designers the ability to innovate and create optimized parts that are too difficult or even impossible to be processed using conventional subtractive fabrication methods [6,7]. Selective laser sintering (SLS) is a common AM technology that uses a high-power laser to sinter small particles of polymer powder into a solid structure based on a 3D model [8]. Its self-supporting ability and capacity for building relatively large parts are some of the major benefits of the SLS process [9,10]. Moreover, can produce durable prototypes and end-use parts with a high dimensional accuracy afforded by the nature of the SLS process [11,12]. SLS is also limited by the raw materials available, even if some polymers with tunable properties have been produced through mineral additives [13,14]. Another limitation of SLS is its poorer mechanical properties compared to traditional manufacturing [15,16], limiting its application for major load-bearing applications [17].

SLS polymers are selected based on the presence of a super-cooling processing window in which there is a large space between the crystallization temperature (T_c_) and melting temperature (T_m_). Therefore, printing at a temperature slightly below T_m_ enables the densification of the SLS powder without reaching its melting point, thus limiting the parts’ distortion. Moreover, by maintaining the temperature above T_c_, the sintered structure remains in an amorphous phase to prevent rapid crystallization, making the powder material more suitable for the production of the final part. Therefore, the parts need to be maintained within the processing window during the build process and slowly cooled down to room temperature to avoid any deformation and crack formation [18,19,20].

Polyamides are the most used polymers in SLS processing, and include Polyamide-11 (PA11), [21,22] Polyamide-6 (PA6) [23,24], and especially Polyamide-12 (PA12) in either its pure or reinforced form [25,26]. The semi-crystalline Polyamide-12 (also called Nylon PA12) accounts for about 95% of the SLS materials used [27] as easy laser sintering can be achieved in comparison with other polymers [28,29]. Table 1 compares the tensile strength for the three different types of polyamides. PA12 is considered a versatile thermoplastic with excellent properties such as toughness, heat, and chemical resistance [30]. Polyamide will behave as a flexible material when thin and as a rigid one when thick. The fabricated parts are usually robust, detailed, and stable for long-term use [31].

SLS is a complex process that usually requires great effort and control in terms of powder and post processing after fabrication to achieve successful printing and high-quality parts. During the SLS process, the material should be kept at an elevated temperature in the build chamber to avoid any deformation of the printouts. The laser provides the necessary energy to exceed the sintering point, making it possible to form the part with the desired geometry [32]. For SLS, part quality and mechanical properties are strongly affected by a large number of printing parameters such as the laser power, laser speed, scan spacing, layer thickness, bed temperature, and build orientation [33,34,35,36]. It is therefore quite important to properly adjust those parameters in order to avoid process instabilities such as a high porosity, which is largely responsible for the poor properties of the tested materials [20]. Starr et al. [37] studied the impact of the process conditions on the mechanical properties of laser-sintered nylon. A high tensile strength was obtained through a high energy density in order to fully melt the applied powder. A higher energy density is required to reach maximum elongation performance, which is more sensitive to the build orientation. The work conducted by Caulfield et al. [12] presented a detailed study about the effects of the energy density level (which comprises the influence of the laser power, hatch spacing, and laser speed) on the mechanical properties of polyamide components. They claimed that using high energy density levels exhibits a more ductile behavior than those obtained at low energy densities. The mechanical test results reported a better elasticity modulus and tensile strength values along the primary x-axis than the secondary z-axis. It was also found that parts built with 0° orientations had a higher ultimate tensile strength and less elongation at fracture relative to the parts with a 90° build orientation.

On the other hand, powder spreading is a crucial step of the SLS process. Controlling the powder quality on the bed affects the quality of the tested parts. The powder should have a good flowability in order to enable the consistent deposition of thin dense layers of powder. Decreasing the porosity content will increase the mechanical properties. The layer thickness of the SLS process is typically between 100–150 μm. Smooth particles with a high sphericity are thus preferable to obtain parts with a desirable microstructure after sintering [38,39] and an adequate surface roughness [40]. The number of crystalline phases in the microstructure also has a significant impact on the mechanical properties of SLS-PA12 parts. Young’s modulus and tensile strength increase with a higher crystallinity, while the elongation at break tends to decrease. The applied process parameters and thermal properties of the material are the major factors determining the amount of crystallinity [41]. Verbelen et al. [42] investigated four commercial polyamide grades by using a new screening methodology that encompasses the complete process chain in laser sintering. They reported that the dilatometry measurements of different PA12 powders showed a reduction in the specific volume during the crystallization phase ranging from 3.9% to 4.7%.

Hofland et al. [20] studied the impact of the process parameters on the mechanical properties by applying Response Surface Methodology (RSM) to analyze the results. They used PA12 powder with a recycled/virgin mixture ratio of 50/50 to produce parts with 0° and 90° build orientations. Dupin et al. [25] compared two types of SLS polyamide 12, Duraform PA (3D systems, Rock Hill, South Carolina, USA) and Innov PA (Exeltec, France), to improve flowability. They used 1% silica in both materials, and it was found that the specimens with the Duraform PA type yielded less porosity than Innov PA, even at a lower energy density.

The present work aims to analyze the impact of the hatch orientation and laser power on the mechanical, microstructural, and morphological properties of 3D-printed PA12 parts. The effect of the heat treatment on the mechanical properties will also be evaluated and compared to the as-built samples

## 2. Materials and Methods

### 2.1. Material and Specimen Preparation

All samples were printed on a P3200HT SLS system from TPM3D (Stratasys company) equipped with a 60W CO_2_ laser. In this study, a Polyamide 12 (PA12) powder (Precimid1171™) from TPM3D with a density of 0.95 g/cm^3^ was used, as it is one of the most widely used materials due to its chemical and physical properties. The chemical structure of PA12 (PA 2200) is shown in Figure 1. Small amounts of fumed silica were added to the PA12 particles to improve powder flowability.

The software ‘’VisCAM RP’’ was used to prepare the build volume and slice models into individual layers before uploading the data to the SLS machine. The main printing parameters used to produce PA12 powder samples are shown in Table 2.

As shown in Figure 2, three orientations were used during parts’ placement on the XY plane in order to study the impact of hatch orientation on mechanical properties. Hatching was conducted by alternating one layer of laser scans at 0° (e.g., parallel to the X-axis) with the following layer at 90° (e.g., perpendicular to the X-axis). As this hatching strategy is applied by the software independently of the part’s orientation on the XY plane, it results in the 0° and 90° orientation parts exhibiting an identical hatching strategy, albeit with a one-layer shift. Due to their hatching similarity, parts with the 0° orientation were placed in the center of the SLS build platform, while parts with a 45° and 90° orientation were positioned all around toward the sides. Three printing runs, with laser power ranging from 45 W to 57 W, were conducted with the purpose of studying the effect of the laser power on the mechanical properties of the 3D-printed parts. The laser power used in this work will be defined as percentage of the maximum laser power of the machine, which is 60 W (LP: 75% is equal to 45 W, LP: 85% is equal to 51 W, and LP: 95% is equal to 57 W).

### 2.2. Size Distribution and Particle Shape

A dynamic image analysis measurement was performed to characterize both the size distribution and particle shape of the PA12 powder used. This analysis was performed using a Camsizer XT equipped with two digital cameras, including one optimized for the analysis of fine particles. Such a setup enables measurement of particles ranging between 2 µm and 8 mm in diameter.

Two PA12 powders were analyzed for comparison: one was the as-received powder, while the second was the un-sintered powder taken from the build volume after only one fabrication. The particle size distribution (PSD) of PA12 powder was identified as a function of percent volume. Furthermore, sphericity was chosen as a shape factor to describe the shape of particles of PA12 powder.

### 2.3. Fourier-Transform Infrared Spectroscopy (FTIR)

Fourier Transform Infrared spectroscopy (FTIR) was used in this work to analyze functional groups of SLS PA12 samples and collect infrared spectra for the structural analysis. This analysis was carried out using a NICOLET™ IS50 attenuated total reflection (ATR) spectrometer. The conditions of measurement were as follows: spectral region of 4000–400 cm^−1^; spectral resolution of 4 cm^−1^.

### 2.4. Differential Scanning Calorimetry (DSC)

Differential scanning calorimetry (DSC) is a common tool used for characterizing materials for laser sintering because it determines the crystallinity and quantifies the melting temperature of printed parts. This analysis was carried out on a 6.6 ± 0.1 mg powder sample using a TA Instruments DSC Q20. The measurements were carried out under a nitrogen atmosphere at a flow rate of 50 mL/min. The crystallinity, X_c_, was calculated using the equation bellow:$$X_{c}\left(\%\right)=\frac{\Delta H_{m}}{\Delta H_{m}^{0}}\times 100$$
where $\Delta H_{m}$ is the enthalpy of fusion and $\Delta H_{m}^{0}$ is the heat of fusion of 100% crystalline PA12, which is taken as 209.3 J.g^−1^ [19].

The annealing process was performed as follows:Heating ramp of 2 °C min^−1^ from room temperature to the annealing temperature Ta (130, 150, and 170 °C);Hold at Ta during the annealing time ta (6 h);Cooling ramp of 2 °C min^−1^ from T_a_ to the room temperature (25 °C);Heating ramp of 10 °C min^−1^ to 220 °C for characterization.

After determination of the appropriate annealing temperature T_a_, sample parts were placed directly in a natural convection oven (Dry-Line series, VWR) for annealing before mechanical testing.

### 2.5. X-ray Diffraction

XRD is a powerful tool used to analyze the atomic or molecular structure of materials. XRD was used here to identify the phase constituent of SLS PA12 powder and samples. Examination of powder and 3D-printed samples was carried out at different laser powers using a XRD X’PERT PRO MPD. Data were acquired over the range of (2θ) 0–90° with a step size of 0.0017 and a scan rate of 7°. min^−1^.

### 2.6. Tensile Test

A tensile test was used to establish tensile properties of 3D-printed SLS-PA12 specimens, including tensile strength, Young’s modulus, and deformation at break. The specimens used were designed according to the ASTM D638-14 “Standard Test Method for tensile properties of plastics”. Three runs were conducted in series to study different laser power and different build orientation as well. Each series comprised six specimens of the D638 type-5 geometry as shown in Figure 3. Testing was carried out on a Criterion C45.105 electromechanical universal testing machine (MTS, USA) equipped with a 10 kN load cell and self-tightening jaws. A crosshead displacement speed of 5 mm min^−1^ was used.

The information obtained by the software were used to enable the calculation of tensile strength, Young’s Modulus, and deformation at break, using the following equations:(1)$$\sigma \left(\mathrm{MPa}\right)=\frac{F\left(N\right)}{s\left({\mathrm{mm}}^{2}\right)}$$
(2)$$E=\frac{\sigma \left(\mathrm{Mpa}\right)}{\varepsilon }$$
(3)$$\varepsilon =\frac{\Delta L}{L0},\;\mathrm{with}\;\Delta L=L-L_{0}$$

### 2.7. Scanning Electron Microscopy (SEM)

Microstructure of both powder and fracture surface was evaluated by scanning electron microscopy (SEM) using a Quanta 200 ESEM (Thermo FEI, Eindhoven, The Netherlands) configured with an EDAX (TSL) EDS/EBSD system for phase identification at high pressures. As-received PA12 powder, used-once SLS powder, and 3D-printed samples cryogenically fractured in liquid nitrogen were coated with a thin layer of electrically conducting gold (Au) to prevent surface charging. Layer arrangement and powder morphology were observed at an acceleration voltage of 5 kV in high-vacuum mode.

## 3. Results and Discussions

### 3.1. Size Distribution, Particle Shape, and Morphology

The particle size distribution (PSD) has a significant impact on the quality of SLS powder, with an ideal diameter between 20 and 80 µm. However, a large number of small diameter particles gives the powder a sticky character that limits its application in the SLS process [8]. Figure 4 illustrates the volume distribution as a function of size for the “As-received powder” and “Powder after fabrication”, where a good PSD can be observed for both powders. Most of the particles for both powders fall in the 30 to 70 μm range, with an additional fraction between 20 to 40 μm. Both powders contain a low number of fine particles, with a diameter of 10 μm or less. The volume distribution for the powder after fabrication exhibits a suitable PSD despite its use in the SLS process, indicating that the powder can be reused after sieving [39]. However, it can be observed that the powder after fabrication exhibits a lesser quantity of particles larger than 60 µm compared to the as-received powder. This could be an effect of the bed-layering process, where larger particles tend to stay above and end up in the overflow bins.

This particle size decrease was confirmed through a comparison of the percentile values (D10, D50 and D90) for both powders (Table 3), where a lower diameter was observed at each percentile in the case of the powder after fabrication. These decreasing diameters imply that the powder contains a higher fraction of smaller particles than at the start. Table 2 also indicates the average particle sphericity for both the as-received powder and the powder after fabrication. It can be observed that the mean value for sphericity does not change significantly because of fabrication, ranging from 0.823 for the as-received powder to 0.818 for the powder after fabrication. While not spherical in shape, the particles for both powders are still considered of a suitable shape for the SLS process.

Figure 5 compares the morphology of the powders in the virgin state (Figure 5a) and after fabrication (Figure 5b). Both samples exhibit particles with a relatively spherical shape, although some elongated particles can be observed. All the particles exhibit a slightly wavy surface texture (similar to cauliflower), which is more pronounced for the particles exposed to the heat cycle of the SLS process. The presence of some satellites is also evident on some of these particles.

### 3.2. Fourier-Transformation Infrared Spectrometry (FTIR)

The ATR-FTIR spectra of the PA12 samples were recorded to provide information about the infrared bands and their roles. Figure 6 displays the spectra of both the as-received PA12 powder and the PA12 powder after fabrication. Table 4 summarizes the different vibrational bands in PA12 and their assignments [43,44,45,46,47,48,49,50,51,52]. Comparing both spectra, it can be observed that the intensities and positions of all bands are almost the same for both PA12 powders. This confirms the lack of influence of selective laser sintering on the chemical composition of PA12 powder.

### 3.3. X-ray Diffraction

The X-ray diffraction patterns of both PA12 powders are shown in Figure 7, revealing their polymorphism and crystalline details. The XRD profile for both powder states shows that polyamide 12 exhibits two characteristic peaks at about 2θ = 20.95° and 21.50°, which are probably characteristics of the α-form of PA12 [14,53]. According to previous studies and the references cited therein, the crystal structure of polyamides has been known to be in the so-called α and γ-forms. The α -form consists of a monoclinic or triclinic lattice with chains in a fully extended planar zigzag arrangement, whereas the γ-form is a pseudo-hexagonal packing of 2_1_ chains. Therefore, PA12 can be crystalized within structures of α and γ phases, where the major γ phase acts as a stable structure [47]. The chains in the α phase are antiparallelly oriented with an extended trans chain conformation, whilst chains in the γ form are oriented parallelly with a twisted helical conformation around the amide groups, making the γ form more stable than the α crystal structure [50].

The patterns relative to the powder after fabrication are almost similar to the patterns for the as-received powder, which confirms the possibility of re-using the powder for SLS after a suitable recycling process.

### 3.4. Differential Scanning Calorimetry

The DSC technique was applied to study the glass transition temperature, melting temperature range, and the degree of crystallinity of the PA12 material. Figure 8 shows the DSC curves for both the as-received powder and the powder after fabrication. Both curves follow almost the same shape, showing three specific thermal transitions: The first transition at around 50 °C is associated with the glass transition (T_g_) phase where the polymer changes to a highly elastic state [54]. The second thermal transition, associated with the melting temperature (T_m_), corresponds to the endothermic peak detected at 182 °C. During cooling, a third transition, observed at 151 °C, is attributed to the crystallization (T_c_) of PA12, where a rearrangement of the molecular chains takes place to create crystalline lamellae inside the continuous amorphous structure. These results show that both types of PA12 powder are in a semi-crystalline state after the cooling process. As shown in Figure 8, there is a large distance between the melting and crystallization peaks, which indicates a tendency of PA12 to warp or curl during the laser-sintering process [42]. This meta-stable thermodynamic region of undercooled polymer is usually called the “SLS sintering window”. It is important to select a sintering temperature in this temperature window range to obtain the best printing results of the material without degrading it. For both powders, the sintering temperature window of PA12 is in the range of [155 °C, 176 °C]. The crystallization temperature (T_c_) must also be avoided as long as possible during processing.

The melting point for the as-received powder is 182.7 °C and is 183.3 °C for the powder after fabrication. The respective fusion heat values of 55.62 J. g^−1^ and 48.49 J. g^−1^ have been measured. After melting, the as-received powder and the powder after fabrication re-solidify with peak crystallization temperatures of 151.9 °C and 149.1 °C, respectively. The solidification (crystallization) temperature is significantly lower than the crystalline melting temperature of PA12; this phenomenon is common in crystalline polymers and is known as super cooling [18]. From Table 5, it can be noted that the as-received PA12 powder exhibit a slightly higher degree of crystallinity than PA12 after fabrication (46.62% against 44.22%). This slight decrease is possibly linked to a reduction in the polymer chain order during the printing process combined with a non-controlled cooling rate of the SLS process after fabrication. The different thermal properties and the crystallization characteristics of the DSC measurements for both powders are all collected in Table 5.

### 3.5. Effect of Laser Power and Hatch Orientation on Tensile Properties

The tensile properties were evaluated at different levels of laser power and hatch orientations and the results are presented in Figure 9. Figure 9a shows the tensile strength (TS) values of PA12′s specimens made in various XY plane orientations when varying the laser power (LP). The measured TS values show the influence of both the XY plane orientation and laser power, with a maximum value of 25.65 MPa achieved at LP: 95% (57W) and a 0° XY plane orientation. These results clearly show that tensile properties rise with an increasing laser power for all part orientations. By increasing the laser power from LP: 75% to LP: 95% in the case of the 0° orientation, the tensile strength is increased from 19.41 MPa to 25.65 MPa. Such an increase can mostly be attributed to an improved coalescence resulting from the higher temperatures achieved by the polymer melt when a higher laser power is used [55]. These results are in good agreement with the findings of Caulfield et al. [12], who claimed that parts built at higher energy density levels (higher laser powers) exhibited a higher tensile strength.

While the effects of laser power on TS are obvious, those linked to hatch orientation are less evident. Such an isolation requires a comparison between the 0° and 90° TS values first, followed by the 45° and 90° TS values. As mentioned previously, the parts made at 0° and 90° XY plane orientation have a nearly identical hatching. This should result in similar TS values as no other SLS parameters differ, something not observed here. Parts made at 90° XY plane orientation exhibit TS values 2 to 10% lower compared to those at 0°, with the greatest difference observed at a low laser power. This decrease in TS for the 90° parts could be explained by their position on the build platen during fabrication relative to the 0° parts. It was shown that there was a need to operate with a powder layer of uniform temperature to achieve builds of multiple parts with similar mechanical properties [27,56,57]. To confirm this, thermal imaging of the preheated powder bed was performed using a Testo 890-2 IR camera. The resulting thermogram, shown in Figure 10, exhibits temperature differences of more than 10 °C between the center and sides of the bed. These gradients could mostly be attributed to a non-uniform heating by the quartz lamps used, possibly because of aging. To highlight this further, the temperature profile along the Y-axis is shown for five locations. As the 0° parts were placed in the center of the build chamber, e.g., in the center third in-between P2 and P4 (Figure 10), they enjoyed a uniform temperature with a variation less than 3 °C. For the 90° parts, their positioning at the periphery entailed more pronounced variations, with some parts partially in regions that were 10 °C lower than the set point of 159 °C. Such a difference could explain their lower TS values against the 0° parts, as the SLS process greatly relies on powder bed heating to supply most of the energy required for sintering particles. Thus, the TS is affected by the powder bed temperature, and the simultaneous production of multiple parts will require a good temperature uniformity to achieve uniform TS values.

The parts made at a 45° XY plane orientation exhibit the overall lowest TS values compared to other orientations made in the same conditions by as much as 7.5%. As these 45° parts were positioned at the periphery, a gradient in the powder bed temperature could partially explain this. As the 45° and 90° parts were interspersed at the periphery, both orientations should exhibit similar TS values, which was not the case here. This difference could be attributed to the hatching orientation. While hatching is conducted alternatively parallel/perpendicular to the applied load in the case of the 90° parts, hatching in 45° parts is performed at an angle relative to the applied load. In a fashion similar to this well-known effect in FDM [58], more loads can be supported along the axis of hatching as it is applied to a continuous string of melted polymer and not at the joining of two strings (or hatches). This could explain the slightly lower TS values obtained for the 45° parts compared to those at 90°. These results confirm that not only laser power and hatch orientation affect the tensile strength of 3D-printed PA12 samples, but that powder bed temperature uniformity must also not be neglected.

Figure 9b displays the Young’s Modulus as a function of laser power for the various XY plane orientations. It is evident that the Young’s modulus values depend on laser power and XY plane orientation, with the best results (1176.6 MPa) achieved at the highest laser power and a 0° angle. This could be explained in part by the laser power and in part by the position on the build platen. Laser power can greatly affect the Young’s Modulus no matter what the part’s angle is, as shown by the 65% increase observed for samples produced at 45° when the laser power was increased from 75 to 95%. This increase in Young’s modulus due to laser power could be attributed to the additional energy applied, which improves the particles’ sintering. This will help achieve better compaction, thus increasing resulting mechanical properties as reported by Singh and al. for polyamide material [59]. Thermal non-uniformity of the powder bed can explain why the 90° parts exhibit a lower Young’s Modulus than the 0° parts, with all other parameters including hatching being similar. The effects of hatching, as seen by comparing the 45° and 90° results, are limited, except when operating at a low laser power. In this instance, the lower energy applied would result in a poor joining of the hatches, which leads to lower mechanical properties.

The Elongation at break as a function of the build orientation and laser power variation was also evaluated (see Figure 9c). However, unlike the TS and Young’s Modulus, the 45° orientation presents the maximum values of elongation (5.13%) for all laser power conditions. In addition, the laser power will affect the elongation at break for parts produced at various XY plane angle. Sintering at a low laser power creates weaker bonds between the powder particles, leading to decreased values of elongation at break [55]. The influence of laser power, while observed in all orientations, is more pronounced for the 45° orientation. At that angle, an increase of 12% in elongation at break was observed when raising laser power from 45W (75%) to 57 W (95%). At similar laser power, the non-uniformity of the powder bed temperature explains the difference in the elongation at break between the parts made at 0° and 90°, while hatching will explain the differences between the 45° and 90° results. In summary, tensile properties are strongly affected by laser power and to a lesser level by the temperature uniformity of the powder bed. The hatching orientation will also affect tensile properties to some extent.

In order to further understand the influence of laser power on the properties of SLS-PA12, the fracture surface of the SLS-PA12 samples was analyzed. Figure 11 shows the SEM observations of the tensile fracture surface for the SLS samples produced at LP: 75% and LP: 95%. The SEM images clearly show particles of PA12 that were melted into the dense part and the presence of some voids in between the layers, especially in the case of LP: 75% (see Figure 11a). These voids favor the delamination of sintered layers, thus explaining the decreased bending strength and inferior rigid behavior. Previous studies have reported that SLS-sintered specimens are porous due to an insufficient heat input that results in their very low mechanical properties [17,46]. The spherical particles observed in Figure 11c,d are un-melted or partially melted PA12 powders. It is clearly evident that those spherical particles are present in large amounts in the case of the sample sintered with a low laser power (LP: 75%) compared to the one sintered at a high laser power (LP: 95%). The PA12 samples with LP: 75% also exhibit a porous interior, with small bonding areas due to the insufficient laser power. This decreases the cohesion between layers while reducing the surface contact between the printed PA12′s layers. The PA12 sample sintered at a 95% LP (Figure 11b) was much smoother, and the fusion effect was improved, though some un-melted powders are still visible in some areas of the cross section (Figure 11d). This adhesive enhancement in the microstructure could explain the observed increase in the tensile properties when higher laser power was used (LP: 95%).

### 3.6. Effect of Laser Power on Thermal and Structural Properties of 3D-Printed Samples

(a)Differential Scanning Calorimetry

A DSC analysis of the printed PA12 parts at different laser powers was performed to determine the impact of this controlling parameter on the thermal characteristics of the PA12 samples manufactured by the SLS process. Figure 12 shows the first heating DSC thermograms of the printed PA12 at various laser powers. From this figure, it is clear that all the samples exhibit the same thermal transitions as powder before fabrication (see Figure 8). It is evident that all the thermograms present a similar thermal behavior and exhibit three thermal transitions. The first heat flow exchange, located at around 45 °C, is associated with the glass transition temperature (T_g_). The second thermal transition, associated with the melting temperature (T_m_), is the endothermic peak detected at 176 °C. During cooling, a third exothermic transition at 146 °C is attributed to the crystallization of PA12, where a re-arrangement of molecular chains takes place to create crystalline lamellae inside the continuous amorphous structure. From these results, it can be concluded that PA12 kept its semi-crystalline property after the cooling process. Table 6 summarizes all characteristic temperatures, associated enthalpies, and the degree of crystallinity of the as-printed samples at various laser powers.

We can notice in Figure 12 a small additional peak at 180 °C in the melting transition. Zarringhalam et al. [41] found the same phenomenon and explained this additional small endotherm as resulting from the unmolten particle core after the laser-sintering process. Generally, the microstructure of SLS parts simultaneously includes fully molten particles and unmolten particle cores surrounded by spherulites. These unmolten powder particles have almost the same melting temperature as the as-received powder, leading to the additional small peak between 180 and 182 °C. The results of the DSC experiments in this study are consistent with previous studies [27,60,61]. The DSC thermograms show the absence of any significant evolution of melting temperature (T_m_), crystallization temperature (T_c_), and glass transition temperature (T_g_) between the printed PA12 at different laser power. It is evident from Table 6 that the variation in laser power produces no significant changes to the degree of crystallinity of samples sintered at various laser power.

However, the degree of crystallization decreased from 46.62% (see Table 4) for the as-received powder to 35.11% for the sample sintered with 75% laser power. This difference could be attributed to a rapid cooling and less energy dissipation during the printing process. Moreover, the decrease in T_g_ from 51.7 to 44.07 °C as a result of sintering correlates strongly with the degree of crystallization of the sample. This can be attributed to the gain in the mobility of the polymer chains as they are not partly anchored inside the crystalline domain.
(b)X-ray Diffraction


Figure 13 displays the profile of the X-ray diffraction patterns of the SLS samples produced at various laser powers (LP: 75%, LP: 85, and LP: 95%). All spectra exhibit similar diffraction peaks, with two major peaks found at 2θ of around 20° and 45°. The peak at 45° appeared after the sintering process; thus, the γ form might be more pronounced even with the presence of the α crystal form [62].

### 3.7. Annealing Impact on Thermal, Structural, and Mechanical Properties

Many studies investigating polymeric materials have considered that using heat treatment post-processing can improve the material properties and crystallinity of SLS parts made from Nylon 12 [63]. These heat treatment (annealing) studies were conducted using different settings of temperature and time. In general, better mechanical properties and crystallinity were obtained when the heat treatment was carried out close to the melting temperature [64]. During annealing, the crystallization processes highly depend on the temperature of the applied annealing procedure. A high temperature would generate an isothermal crystallization process, in which the non-crystalline polymer chains have enough energy to form more crystalline regions and an optimal arrangement [65,66]. In this work, the choice of annealing temperatures was based on this latter theory.

A DSC analysis was used to optimize the annealing temperature by determining the adequate cycle yielding the highest thermal property and degree of crystallinity. The printed parts with the selected optimized printing parameters ([0°] orientation and 95% laser power) were annealed for six hours at various temperatures (130 °C, 150 °C, and 170 °C) to allow for the relaxation of the residual stress generated during their printing process. The annealed parts were analyzed by DSC, using the first heating cycle to characterize the thermal history experienced during the annealing process. During this heating cycle, the material achieved its melting temperature (T_m_) to characterize the melting enthalpy, and then the degree of crystallinity (X_c_) generated during the annealing cycle. Figure 14 shows the first heating and cooling DSC thermograms of the unannealed and annealed samples under various annealing cycles. The X_c_ and all thermal transitions were measured (see Table 6) for the samples prepared by cutting a small amount of material from the tensile test specimens (before the tests were carried out).

Table 7 summarizes the DSC results for various parameters such as T_g_, T_m_, and X_c_. From this table, the DSC results show that the T_g_ of the annealed parts shows a shift to higher temperatures with the increasing annealing temperature. When the annealing temperature was increased from 130 to 170 °C, the T_g_ increased from 40.8 to 50.3 °C. Even though the maximum annealing temperature was achieved (170 °C), the T_g_ value was still higher than the unannealed printed parts. It can also be noted that raising the annealing temperature from 130 to 170 °C results in an increase in the heat flow of melting from 65.51 to 76.51 J.g^−1^. An increase in the relative degree of crystallinity was also observed, from 31.29% to 36.55%. This latter crystallinity enhancement could be the result of a phenomenon called secondary crystallization, which increases the lamellar form of PA12. Moreover, the increase in T_g_ because of annealing correlates strongly with the degree of crystallinity of the printed PA12′s parts. Annealing thus induces strong intermolecular interactions between polymer chains. Thus, this increase is attributed to the loss of mobility of polymer chains as they are partly anchored inside the crystalline region.

The annealed specimens were then subjected to mechanical testing to record the Young’s modulus, tensile strength, and strain at break. Figure 15 shows the typical tensile stress–strain curves of unannealed and annealed printed PA12 parts according to the various annealing conditions. All tested specimens exhibit a maximum of the stress/strain curve, followed by brittle deformation. The results indicate that tensile stress and its strain increases when the annealing temperature is increased from 130 to 170 °C. The annealed parts at 170 °C show the maximum tensile stress and strain compared to the unannealed samples. These results indicate the fact that heat treating the PLA12 parts at 170 °C for 6 h allows the material sufficient time for crystallization and the re-arrangement of the polymer chains.

From Figure 16, it can be observed that annealing PA12 material favorably affects the tensile properties of the printed PA12 parts. For untreated samples, the Young’s modulus and tensile strength exhibit the lowest values (1176.7 MPa and 25.5 MPa, respectively) compared to the annealed samples at 170 °C (1276.23 MPa and 29.2 MPa respectively) (see Figure 16a). This change in rigidity and strength is related to changes in the microstructures of the material, as the annealed samples exhibit a higher degree of crystallinity and glass transition temperature as discussed above. These results are in good agreement with the work of Liu et al. [67], who reported that high-temperature annealing (173 °C) can remarkably enhance the mechanical strength of printed PA12 specimens. The work by Zarringhalam and al. [64] confirmed that using heat treatment as a post-processing technique can improve the tensile strength and Young’s modulus of SLS parts made from Nylon 12. However, as shown in Figure 16b, annealing at 170 °C/6 h leads to an increase in ductility where the elongation at break increases to 5.66%, while the as-printed (un-annealed) samples exhibit a value of 4.07%.

From the above results, it has been confirmed that high-temperature annealing (170 °C) yields the best improvement in Young’s modulus (+9 MPa, or ~8.4%), tensile strength (+3.7 MPa, or ~14.5%), and elongation at break (+1.59% MPa, or ~39%) over the unannealed parts. It can be concluded from these results that annealing had a higher percent contribution to the mechanical performance over the duration of annealing. This confirms the importance of the annealing process for achieving proper chain crystallization, thus enhancing the mechanical properties of 3D parts [58].

## 4. Conclusions

This study evaluated the effect of laser power and hatch orientation on the tensile properties and morphology of the SLS PA12-produced parts. The main objective was to identify the suitable laser power and hatch orientation leading to better mechanical properties and high-quality parts. Different methods were used to study the SLS parts by considering the morphological, structural, and mechanical properties using XRD, FTIR, DSC, tensile testing, and SEM characterizations.

The results confirmed the significant impact of laser power, while the effects of hatching were almost unnoticeable when using a high laser power. A more significant condition is the uniformity of the powder bed temperature, a factor that is seldom considered. This needs to be accounted for because of its effects on the mechanical properties. However, the operator has little recourse with respect to these conditions, which are strongly dependent on the quality of the SLS system.

Operating at a high laser power minimized the presence of spherical particles normally related to un-melted powder and yielded an improved microstructure. It was also observed that reducing the laser power to LP: 75% decreases the mechanical properties, with the parts exhibiting spherical particles and a poor microstructure. Heat treating SLS-produced PA12 parts showed the positive impact of annealing, especially at 170 °C, on the tensile properties. This can be related to changes in the microstructure of the PA12 parts.

## Figures and Tables

**Figure 1 polymers-14-03674-f001:**
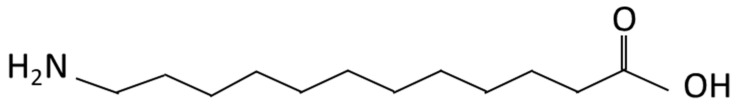
Chemical structure of Nylon (PA12), polydodecanolactam.

**Figure 2 polymers-14-03674-f002:**
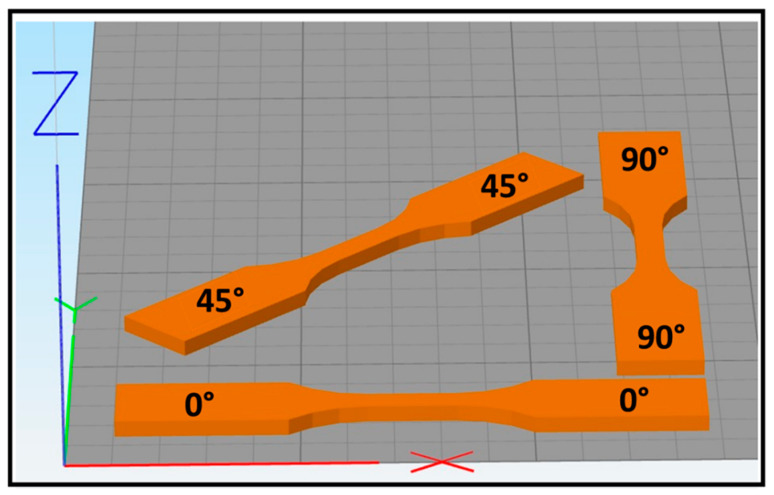
Illustration of different specimen orientations.

**Figure 3 polymers-14-03674-f003:**
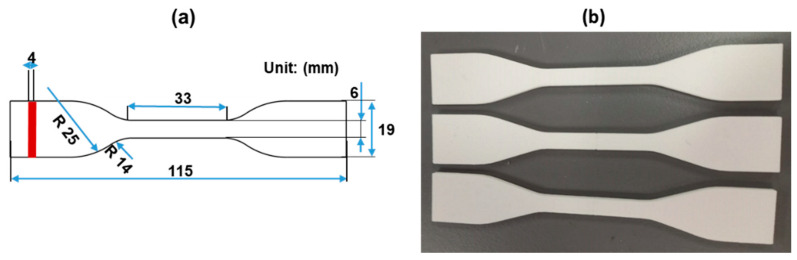
(**a**) tensile test bar dimensions; (**b**) SLS-printed specimens for tensile test.

**Figure 4 polymers-14-03674-f004:**
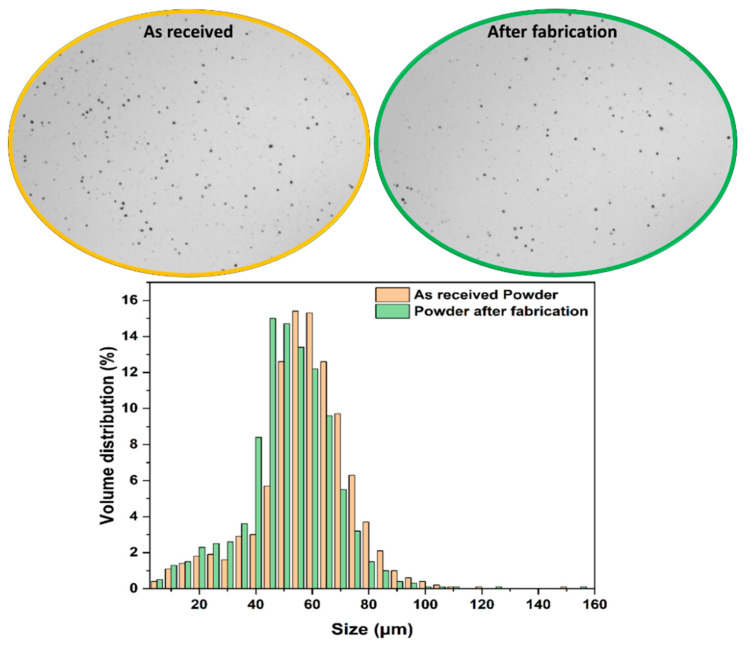
Volume distribution as function of the size for the “As-received powder” and “Powder after fabrication”.

**Figure 5 polymers-14-03674-f005:**
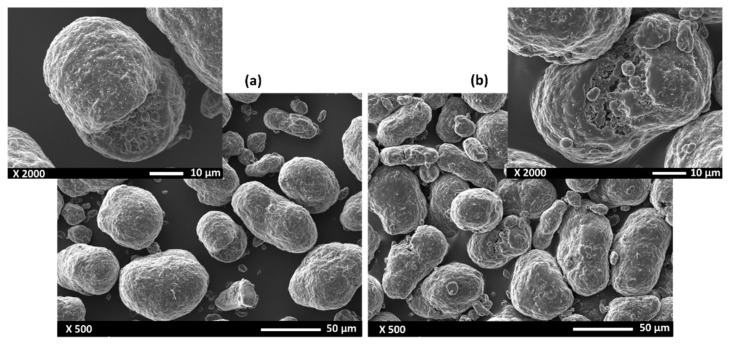
SEM micrographs of the PA12 powders: (**a**) the as-received powder; (**b**) powder after fabrication.

**Figure 6 polymers-14-03674-f006:**
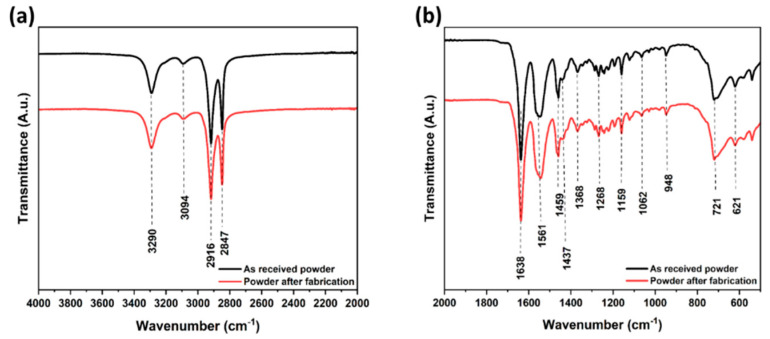
Fourier-transform infrared (FTIR) spectra of PA12 powders: As-received powder; Powder after fabrication. (**a**)-wavenumber from 2000 to 4000 cm^−1^; (**b**)-wavenumber from 550 to 2000 cm^−1^.

**Figure 7 polymers-14-03674-f007:**
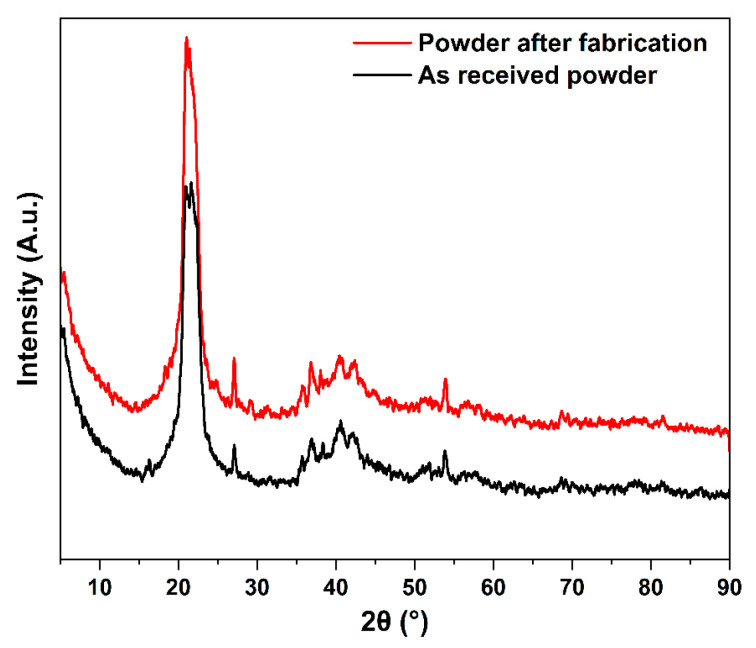
X-ray diffraction patterns of the as-received powder and powder after fabrication in range 4.5–90°.

**Figure 8 polymers-14-03674-f008:**
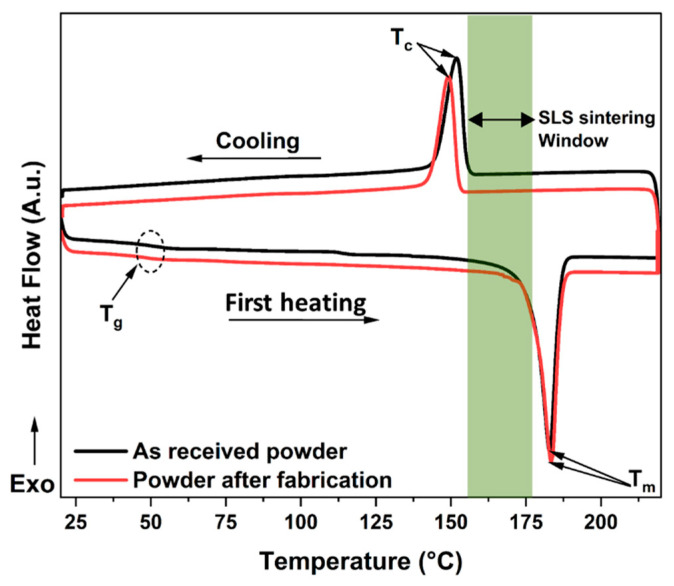
DSC thermograms of PA12 powders: as-received powder and powder after fabrication with the SLS sintering window shown in light blue.

**Figure 9 polymers-14-03674-f009:**
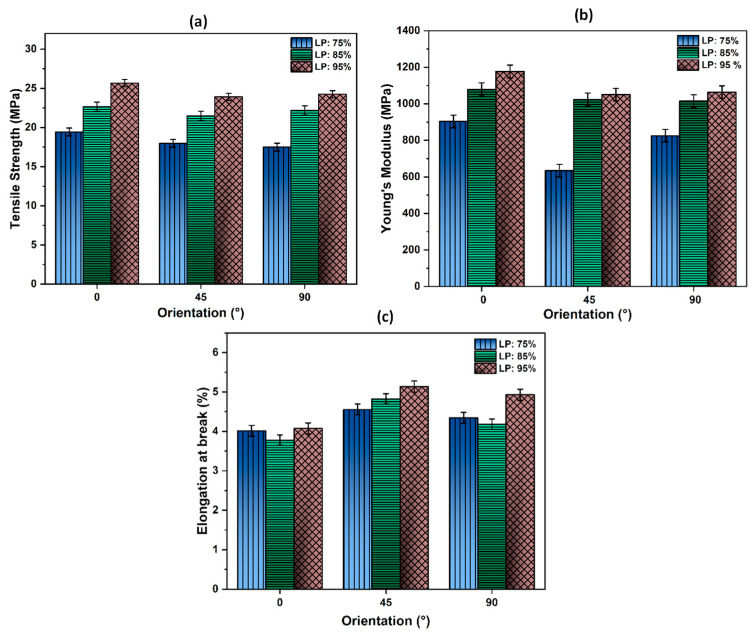
Tensile properties as a function of XY plane orientation and laser power: (**a**) Tensile Strength; (**b**) Young’s Modulus; (**c**) Elongation at break.

**Figure 10 polymers-14-03674-f010:**
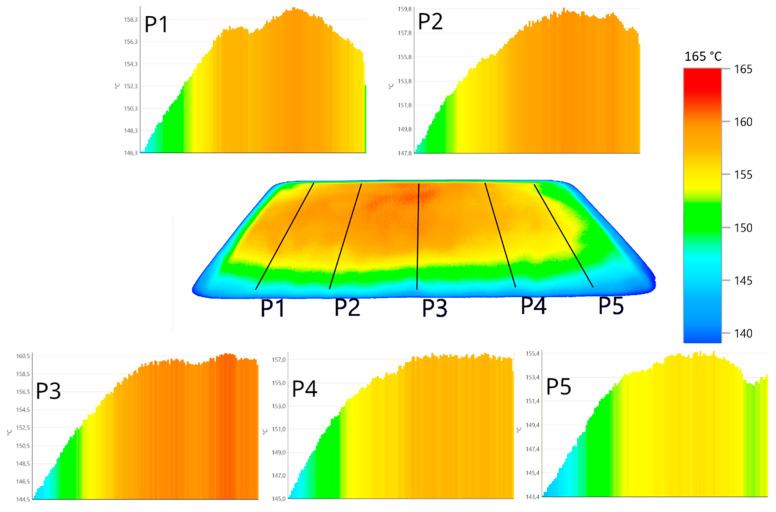
Thermal imagery of the SLS powder layer during preheating. P1 to P5 are front-to-back temperature profile taken from left to right of the platen (note that the Y-axis does not have the same numeral scale).

**Figure 11 polymers-14-03674-f011:**
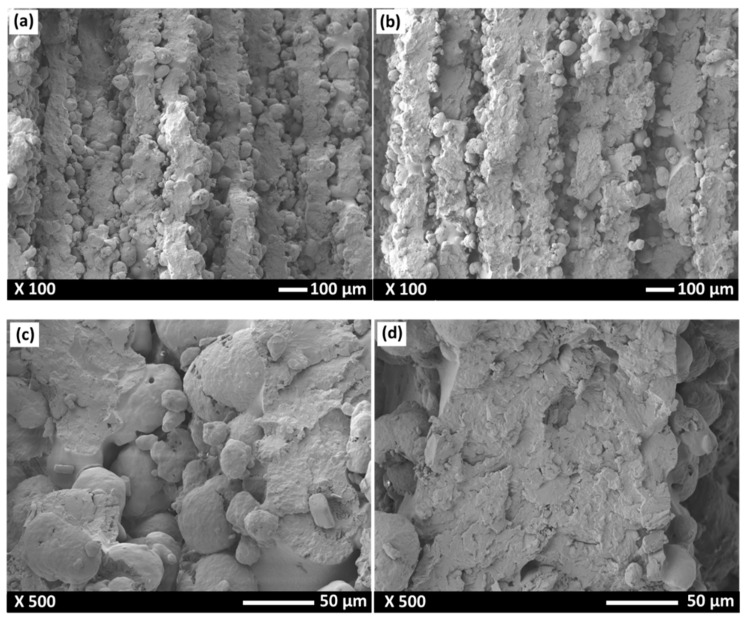
SEM micrographs of the fractured surfaces of SLS-printed PA12 specimens: (**a**) LP: 75% [×100], (**b**) LP: 95% [×100], (**c**) LP: 75% [×500], and (**d**) LP: 95% [×500].

**Figure 12 polymers-14-03674-f012:**
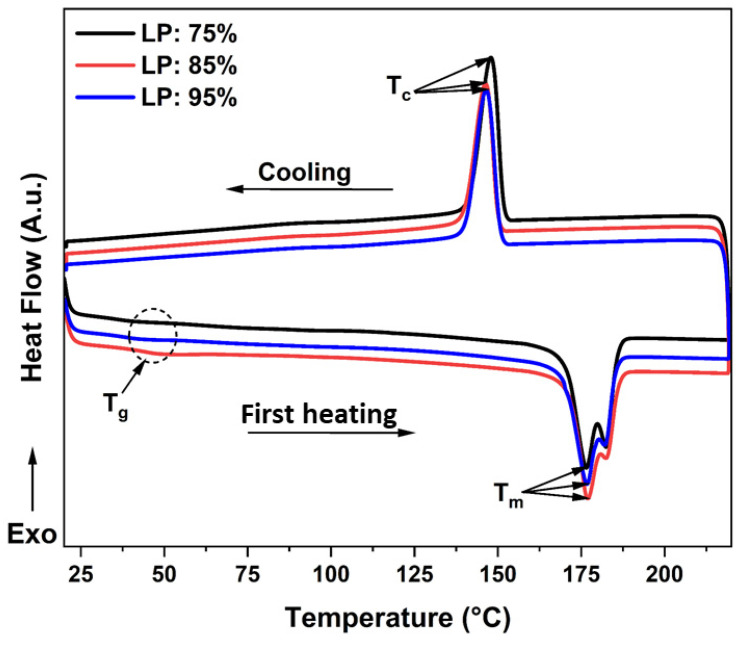
DSC thermograms of PA12 samples with the three laser powers used: LP-75%, LP-85%, and LP-95%.

**Figure 13 polymers-14-03674-f013:**
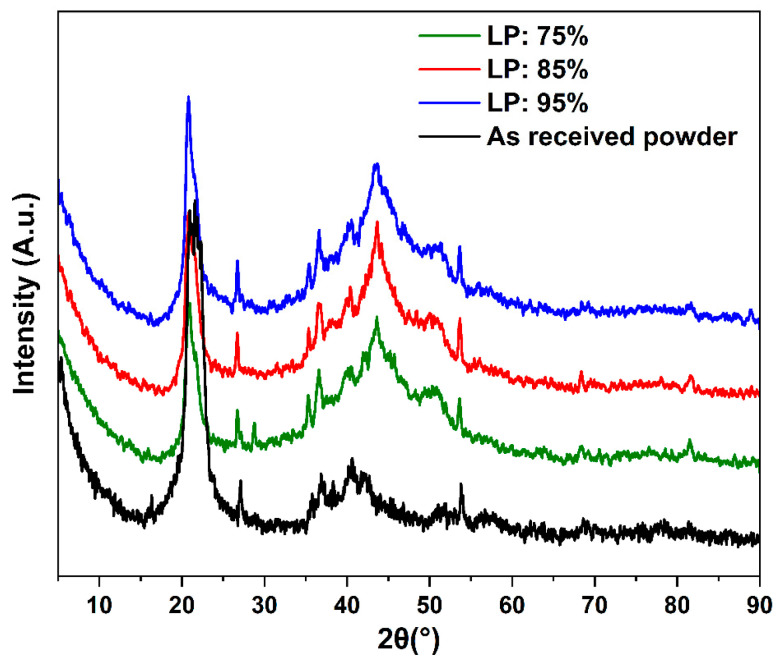
X-ray diffraction patterns of the SLS samples parts with different laser powers in range from 4.5–90°.

**Figure 14 polymers-14-03674-f014:**
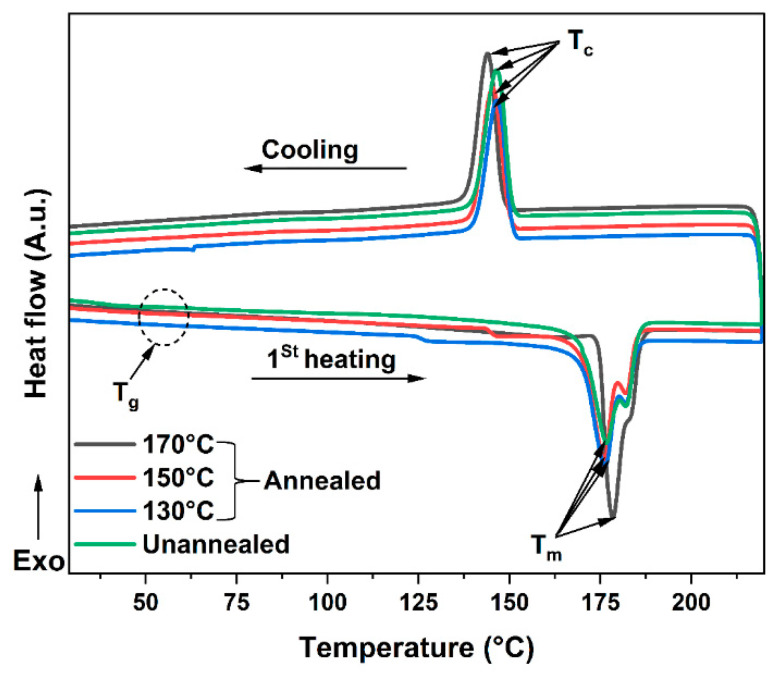
Differential scanning calorimetry curves for unannealed and annealed PA12′s samples.

**Figure 15 polymers-14-03674-f015:**
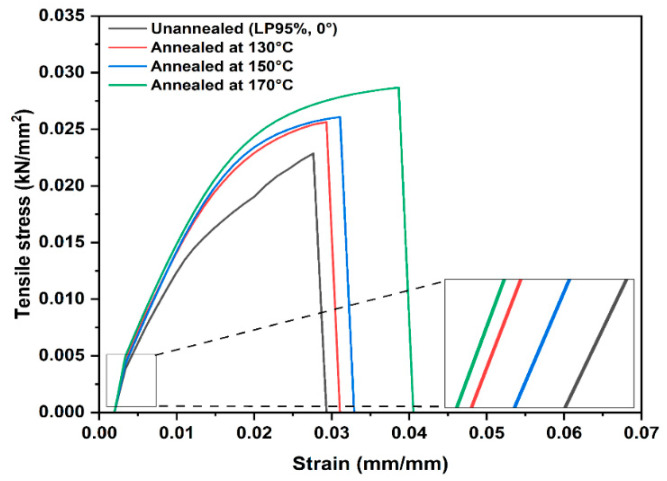
Stress–strain curves of printed PA12 in different conditions. Unannealed and annealed at: 130, 150, and 170 °C.

**Figure 16 polymers-14-03674-f016:**
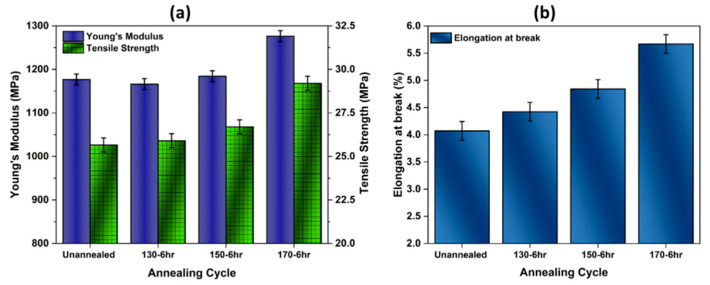
Effect of annealing on tensile properties of PA12 samples’ parts: (**a**) results of Young’s modulus and tensile strength; (**b**) results of elongation at break.

**Table 1 polymers-14-03674-t001:** Tensile strength of some types of polyamides with their references.

Material	PA6	PA11	PA12
**Tensile strength (MPa)**	3.75	52	26.25
**References**	[23]	[21]	[29]

**Table 2 polymers-14-03674-t002:** Main printing parameters used in selective laser sintering of PA12 powder.

Parameters	Values	Units
**Laser power**	75–85–95	(%)
**Part orientation (XY plane)**	0–45–90	(°)
**Layer thickness**	0.15	(mm)
**Platform temperature**	169	(°C)
**Chamber temperature**	135	(°C)
**Moving plate temperature**	140	(°C)
**Hatch spacing**	0.220	(mm)
**Diameter of laser beam**	0.22	(mm)
**Infill**	100	(%)
**Scanning speed**	13	(mm.s^−1^)
**Hatch orientation (XY plane)**	0–90–0–90	(°)

**Table 3 polymers-14-03674-t003:** Powder sample characteristics.

Material/Characteristics	As-Received Powder	Powder after Fabrication
**D_10_ (μm)**	33.2	28.8
**D_50_ (μm)**	55.7	49.1
**D_90_ (μm)**	73.3	66.7
**Mean value Sphericity**	0.823	0.818

**Table 4 polymers-14-03674-t004:** Characteristic infrared bands and their assignments of SLS PA12 Powder.

Vibrational Frequency [cm^−1^].	Assignments
**3290**	*ʋ* (N–H) stretching
**3094**	Fermi resonance of *ʋ* (N–H) stretching
**2916**	*ʋ* (CH_2_) asymmetric stretching
**2847**	*ʋ* (CH_2_) symmetric stretching
**1638**	Amide-I (*ʋ* (C=O) stretching and *ʋ* (C–N) stretching)
**1561**	Amide-II (*δ* (N–H) bending and *ʋ* (C–N) stretching)
**1459**	*δ* (CH_2_) scissoring
**1368**	*δ* (CH_2_) twisting
**1268**	Amide-III (*ʋ* (C–N) stretching and *δ* (C=O) in-plane bending)
**1159**	Skeletal motion CO–NH
**1062**	Skeletal motion CO–NH
**948**	*δ* (CO-NH) in-plane bending
**721**	*ρ* (CH_2_) rocking
**621**	Amide-IV (*δ* (N–H) out-of-plane bending)

**Table 5 polymers-14-03674-t005:** DSC data corresponding to the first heating–cooling scan for the as-received powder and powder after fabrication.

Powder State	T_g_ (°C)	T_m_ (°C)	T_c_ (°C)	∆H_m_ (J.g^−1^)	∆H_c_ (J.g^−1^)	$X_{c}\;(\%)$
**As-received powder**	51.7	182.7	151.9	97.59	55.62	46.62
**Powder after fabrication**	49.0	183.3	149.1	92.67	48.49	44.27

**Table 6 polymers-14-03674-t006:** DSC data corresponding to the first heating–cooling scan for the different laser powers used.

Specimen	T_g_ (°C)	T_m_ (°C)	T_c_ (°C)	∆H_m_ (J.g^−1^)	∆H_c_ (J.g^−1^)	$X_{c}\;(\%)$
**LP: 75%**	44.07	176.54	147.89	73.50	56.04	35.11
**LP: 85%**	45.45	176.93	146.30	67.50	50.94	32.25
**LP: 95%**	46.14	176.73	146.47	69.31	54.50	33.11

**Table 7 polymers-14-03674-t007:** DSC data corresponding to the first heating–cooling scan for the different annealing temperature used.

Specimen State	T_g_ (°C)	T_m_ (°C)	T_c_ (°C)	∆H_m_ (J.g^−1^)	∆H_c_ (J.g^−1^)	$X_{c}\;(\%)$
**Unannealed**	46.1	176.7	146.4	69.31	54.50	33.11
**Annealed at 130 °C/6 h**	40.8	176.3	146.5	65.51	49.78	31.29
**Annealed at 150 °C/6 h**	44.4	176.0	145.3	69.66	50.09	33.28
**Annealed at 170 °C/6 h**	50.3	178,4	143.9	76.51	51.79	36.55

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
