# Peer review of "Effects of Laser Power and Hatch Orientation on Final Properties of PA12 Parts Produced by Selective Laser Sintering"

_polymers, 2022, doi:10.3390/polym14173674_

Round 1

Reviewer 1 Report

The paper titled “Effects of Laser Power and Build Orientation on Final Properties of PA12 Parts Produced by Selective Laser Sintering”, by Anouar El Magri et al. is, in my opinion, not sufficiently sound for publication in the journal. The following suggestions should be considered:

1. More details about the Polyamide 12 powder should be provided in Materials and Methods, such as molecular weight and the type of powder in TPM3D.

2. “the second is powder taken from the build volume after only one fabrication.” Is the second kind of powder the sintered powder or the unsintered powder around the build volume? Please describe clearly. If it is the former, how does the author obtain the powder from the build volume without affecting the particle size of the powder?

3. Hatch spacing should be added to Table 1. Main printing parameters used in selective laser sintering of PA12 powder.

4. The author constructed PA12 samples with laser power of 75%, 85% and 95%, respectively, and found that the tensile strength of the sample with 95% laser power was the best. With the further increase of laser power, will the tensile strength of the sample continue to improve?

5. Why using an angle of 0° produced samples with higher Young’s Modulus over other build orientations? The reasons for the better results require scientific explanation.

6. Why is the annealing time chosen to be 6 hours? Does the annealing time also affect the degree of crystallization of the sample?

7. In Figure 14, The unannealed sample seems to have a greater elongation at break than samples annealed at 130 ℃ and 170 ℃. This does not match the results in Figure 15(b). Could the author clarify this doubt?

8. This is true for the using of SLS to fabricate polymer with controllable properties, see e.g. Bioactive Materials, 2021, 6:490-502; Polymer Testing, 2016, 53: 217-226. This important information should be integrated into the paper.

Author Response

Authors thank you for your comments and suggestions

Reviewer 2 Report

The manuscript entitled Effects of Laser Power and Build Orientation on Final Properties of PA12 Parts Produced by Selective Laser Sintering is worth of researchers interest, but carefully revision is required.

Some shortcomings are listed bellow:

-PA12 that is an abbreviation, is not explained. It has to be explained at first usage. It is introduced into Abstract and main text without any details. Please introduce the IUPAC name of the commercially known polyamide 12.

-The same action for SLS.

-References lack to be uniformly written and according to the journal rules. Please revise carefully. Major part of the references are old (recent publications should be cited) and not all authors are listed (for instance [28], [34], [37]).

-In Introduction the comparison of polyamides with other polymers should be given in more details. A table comparing the best properties/performances of the polymers suitable for this process (and the references column) is demanded to clarify your approach.

-Your purpose and the novelty of your approach should be clearly emphasized.

-Table 3 is doubled by text. Please reduce the explaining text to the main necessity.

_ Please clearly and in detail explain the elongation values and what are the optimum values expected.

- Revise the Conclusions section and specify the main results and the advancements of this work.

Author Response

(The authors gave the same response as above.)

Reviewer 3 Report

This paper explores the effect of two key SLS parameters, laser power and build direction, on the tensile, structural, thermal, and morphological properties of fabricated PA12 parts. Both of these evaluation parameters have an impact on the final performance of the printed PA12 part. The best factor levels were achieved at 95% laser power and 0° build orientation. Parts produced using the optimized SLS parameters are then annealed to relieve residual stress and increase crystallinity. The results showed that annealing the SLS part at 170°C for 6 hours significantly improved the thermal, structural and tensile properties of the 3D printed PA12 part. I considered it can be published in Iranian Polymer Journal after a minor revision:

1.     As described in the article: “the major γ phase acts as a stable structure”

Why there are mainly characteristic peaks of α phase in the XRD peaks of the two powders?

Why choose the three build orientations of 0°, 45°, and 90° for the three build orientations of the specimen? What is the principle?

2.       Why does the TS value decrease with increasing build orientation, from 0° to 45° and 90°? Please explain.

3.       “The increase in Tg because of annealing correlates strongly with the degree of crystallinity of the printed PA12’s parts”

4.     What is the relationship between “increase in Tg” and “the degree of crystallinity of the printed PA12’s parts”, please explain in detail. add some references such as “Journal of Cleaner Production, 359 (2022) 132134., DOI: 10.1007/s42114-022-00458-7.”.

Author Response

(The authors gave the same response as above.)

Round 2

Reviewer 1 Report

The paper can be accepted!

Reviewer 2 Report

The manuscript was carefully revised and completed, so I agree to be published.

Reviewer 3 Report

The author has well replied all my questions and revised their manuscript according to my suggestions. I accept publish this manuscript.